# OPEN-SET 3D DETECTION VIA IMAGE-LEVEL CLASS AND DE-BIASED CROSS-MODAL CONTRASTIVE LEARNING

## ABSTRACT

Current point-cloud detectors have difficulty in detecting the open-set objects in the real world, due to their limited generalization capability. Moreover, collecting and fully annotating a point-cloud detection dataset with a large number of classes of objects is extremely laborious and expensive, leading to the limited class size of existing point-cloud datasets and hindering the model from learning general representations for open-set point-cloud detection. Instead of seeking well-annotated point-cloud datasets, we resort to ImageNet1K to broaden the vocabulary of point-cloud detectors. Specifically, we propose OS-3DETIC, an **O**pen-**S**et **3**D **DET**ector using **I**mage-level **C**lass supervision. Intuitively, we take advantage of two modalities, the image modality for recognition and the point-cloud modality for localization, to generate pseudo-labels for unseen classes. We then propose a novel de-biased cross-modal contrastive learning strategy to transfer knowledge from image to point-cloud. Without hurting the latency during inference, OS-3DETIC makes the well-known point-cloud detector capable of achieving open-set detection. Extensive experiments demonstrate that the proposed OS-3DETIC achieves at least 10.77% mAP improvement (absolute value) and 9.56% mAP improvement (absolute value) by a wide range of baselines on the SUN-RGBD and ScanNetV2, respectively. Moreover, we conduct sufficient experiments to shed light on why the proposed OS-3DETIC works.

## 1 INTRODUCTION

3D point-cloud detection is defined as finding objects (localization) in point-cloud and naming them (classification). Recently, deep learning based 3D detectors have achieved significant progress. However, most methods are developed on point-cloud detection datasets with limited classes, whereas the real world has a cornucopia of classes. It is common for 3D detectors to encounter objects that had never occurred during training, resulting in failure to generalize to real-life scenarios. Therefore, it is extremely important to design an open-set point-cloud detector which is able to generalize to unseen classes. The key ingredient of open-set detection is that the model learns sufficient knowledge thus is able to output general representations. To achieve this, in the image field, typical open-set classification and detection either require to introduce large-scale image-text pairs or image datasets with sufficient labels. For example, CLIP Radford et al. (2021) introduced 400 million image-text pairs for pre-training to help visual models learn general representation. Detic Zhou et al. (2022) leverages ImageNet21K to extend the knowledge of image detectors. OVR-CNN Zareian et al. (2021) uses the language pretrained embedding layer to broaden the vocabulary of the 2D detector.

However in the point-cloud field, as far as we know, there are few studies for the open-set point-cloud detection. The most notable hindrance is that we can hardly obtain large-scale point-cloud data and labels (or optionally, captions like the aforementioned image field), due to the difficulty of collection and annotation. The scarcity of point-cloud data and labels drastically restricts the point-cloud model from learning sufficient knowledge and obtaining general representations. Therefore, this limitation motivates us to ask - *Can we transfer the knowledge from images to point-cloud so that the point-cloud model is capable of learning general representations?*

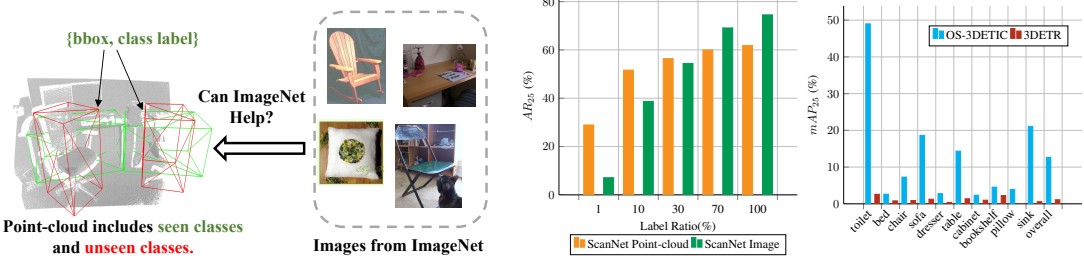

(a) ImageNet helps open-set 3D point-cloud detection.

(b) $AR_{25}$ w.r.t label ration on the ScanNet dataset.

(c) $mAP_{25}$ w.r.t categories on the ScanNet dataset.

Figure 1: (a). The left point-cloud includes the tables (green) and the chairs (red). The tables are labeled and denoted as seen objects, while the chairs are unseen objects. Right part shows examples from ImageNet1K which has sufficient labels. We aim at utilizing large-scale ImageNet1K to broaden the vocabulary of point-cloud detector. (b). We train the 3DETR Misra et al. (2021) and DETR Carion et al. (2020) on the ScanNet dataset. Under low-data regime, e.g., with randomly sampling 10 % data, $AR_{25}$ is still at a high-level accuracy, even competitive to using 100 % data, which demonstrates that localization in point-cloud object detection generalizes well. (c). The performance comparison between the proposed OS-3DETIC and the baseline 3DETR on the ScanNet dataset. OS-3DETIC significantly improves the baseline by a large margin on all the categories.

Our work shows that the answer is positive. Intrinsically, images are dense RGB pixels while point-clouds consist of sparse xyz points. Although the large gap exists, both point-cloud and image are visual representations to the physical world and can express the same visual concepts Xu et al. (2021a); Park et al. (2022). Human-beings have no problem understanding both modalities. But two issues still exist: what kind of image data can we use and how to use the image data?

It is straightforward that we directly utilize the image detection dataset with fully bounding boxes and class labels, and transfer the knowledge from the 2D detector to 3D. However, bounding-box level annotation is still laborious and difficult to scale, while open-set detection requires rich labels to help the detector learn sufficient knowledge Zhou et al. (2022). Therefore, in this paper, instead of seeking to build a large-scale point-cloud dataset or use 2D detection datasets, we open-up another path by resorting to large-scale image datasets with image-level class supervision, ImageNet1K Krizhevsky et al. (2012), to enable the point-cloud detector capable of learning general representations, thus broadening the vocabulary of the point-cloud detector, as shown in Fig. 1(a).

Open-set 3D detection it to detect unseen categories without corresponding 3D labels. Note that, in this setting, open-set is defined in terms of the 3D detection, we can make use of external knowledge such as ImageNet, which only provides Image-level category knowledge. Specifically, the proposed OS-3DETIC makes the 3D detector learning sufficient knowledge from image-level supervision thus achieving open-set point-cloud detection, and it is a synergy of two components: 1) Make full use of knowledge learning from ImageNet and generalizability of localization on point-cloud to generate pseudo-labels for unseen class. 2) We design a de-biased cross-modal contrastive learning with distance-aware temperature to capture the shared low-dimensional space within and across modalities, thus better transferring the sufficient knowledge from image domain to point-cloud domain. It is noteworthy to mention that during training, we introduce the paired images to narrow the gap between the point-cloud data and the images from ImageNet, but we do not need any extra annotations except the Lidar-Camera transformation matrix.

Extensive experiments show that OS-3DETIC outperforms a wide range of state-of-the-art baselines by at least 10.77% mAP (absolute) and 9.56% mAP (absolute) without hurting the latency of the original 3D detector, on the unseen classes of SUN RGB-D Song et al. (2015) and the ScanNet Dai et al. (2017), respectively. An example on the ScanNet dataset is shown in Fig. 1(c). Sufficient ablation studies shed light on why the OS-3DETIC works. Overall, our contribution is as follows:

- We propose an open-set 3D detector with image-level class, termed as OS-3DETIC, which is a synergy of two components: the pseudo-label generation from two modalities, and the de-biased cross-modal contrastive learning with distance-aware temperature.

- OS-3DETIC can be regarded as a superior baseline in open-set 3D point-cloud detection.
- Extensive experiments demostrate the effectiveness of OS-3DETIC, and we also provide sufficient analysis to uncover why it works.

## 2 RELATED WORK

open-set object detection targets to detect novel classes that are never provided labels during the training Bansal et al. (2018); Gu et al. (2021); Rahman et al. (2020a;b); Zhou et al. (2022); Zareian et al. (2021). The most similar work to us is Detic Zhou et al. (2022), which utilizes ImageNet21K to broaden the classifier of the 2D detector. Yet, it is infeasible to directly use the same method to broaden the classifier of the point-cloud detector, due to the large gap between image and point-cloud. Different from Detic that transfers knowledge within the same modality, we propose to transfer knowledge from ImageNet to a totally different modality. Point-Cloud Detection has been actively researched Yang et al. (2018); Chen et al. (2017); Wu et al. (2018; 2019); Xu et al. (2020); Shi et al. (2019); Xu et al. (2018; 2021b); Zhou & Tuzel (2018); Yan et al. (2018); Shi et al. (2020b;a); Dosovitskiy et al. (2020); Wu et al. (2020); Liu et al. (2021c); Misra et al. (2021); Zhao et al. (2021), *yet to the best of our knowledge, open-set 3D point-cloud detection has not been widely explored, existing methods Cen et al. (2021; 2022); Wong et al. (2020) mainly focus on identify the unknown objects from known one, whereas in our case, we further assign a name to each unknown object, which is similar to Xu et al. (2021a); Zhang et al. (2021); Cheraghian et al. (2019b;a; 2021), but these works mainly study classification.*

## 3 METHOD

### 3.1 NOTATION AND PRELIMINARIES

We use $\mathbf{I} \in \mathcal{R}^{3 \times H \times W}$ to represent image, and $\mathbf{P} = \{\mathbf{p}_i \in \mathcal{R}^3, i = 1, 2, 3..., N\}$ to represent point-cloud, where $N$ is the point number in the point-cloud. During training, we use 1) point-cloud dataset denoted as $\mathcal{D}^{pc} = \{(\mathbf{P}, (\mathbf{b}_{3D} \in \mathcal{R}^7, \mathbf{c}_{3D})_k)_j\}_{j=1}^{|\mathcal{D}^{pc}|}$, with vocabulary size $\mathcal{C}_{pc}$, where $\mathbf{b}_{3D}$ is the annotation of 3D bounding box, $\mathbf{c}_{3D}$ is the corresponding classification label; 2) paired image dataset denoted as $\mathcal{D}^{img} = \{\mathbf{I}_j\}_{j=1}^{|\mathcal{D}^{img}|}$; 3) ImageNet1K dataset denoted as $\mathcal{D}^{ign} = \{(\mathbf{I}, \mathbf{c}_{2D}^{ign})_j\}_{j=1}^{|\mathcal{D}^{ign}|}$, with vocabulary $\mathcal{C}_{ign}$, where $\mathbf{c}_{2D}^{ign}$ denotes classfication label of the image in ImageNet1K. During test, we evaluate on the vocabulary $\mathcal{C}_{test}$, where $\mathcal{C}_{ign} \geq \mathcal{C}_{test} > \mathcal{C}_{pc}$.

A typical point-cloud detector deals with localization and classification, where the localization module outputs bounding boxes $\hat{\mathbf{b}}_{3D} \in \mathcal{R}^7$ and we could get its corresponding point-cloud ROI features $\mathbf{f}_{3D}$. Similarly, we can project the 3D bounding box into 2D image via projection matrix $K$, i.e., $\hat{\mathbf{b}}_{2D} \in \mathcal{R}^4$ and index the corresponding image ROI feature $\mathbf{f}_{2D}$. Then we can use the $\mathbf{f}_{3D}$ and $\mathbf{f}_{2D}$ to predict the class of each object.

### 3.2 OS-3DETIC: OPEN-SET 3D DETECTOR WITH IMAGE CLASSES

We use ImageNet1K to broaden the vocabulary of the point-cloud detector. To transfer the knowledge contained in ImageNet1K to the point-cloud detector, paired image is introduced to bridge these two modalities. Specifically, We design a two-phase training strategy to enable open-set point-cloud detection. The first-phase training is similar to Detic Zhou et al. (2022), which aims at leveraging ImageNet to help the 2D detector be able to learn sufficient knowledge. The second-phase is the core of the proposed OS-3DETIC, which aims at transferring the knowledge of the 2D detector to the 3D detector via customized pseudo-label strategy and de-biased cross-modal contrastive learning, as shown in Fig. 2. During inference, we only need the 3D detector without any extra models or modalities. OS-3DETIC is developed on top of 3DETR Misra et al. (2021), and we use DETR Carion et al. (2020) as the 2D detector. The details are illustrated below.

In the first stage, the point-cloud $\mathbf{P}$ in $\mathcal{D}^{pc}$ is injected to the 3D detector, which is supervised by the ground truth $\{\mathbf{b}_{3D}, \mathbf{c}_{3D}\}$ of seen objects. The paired images $\mathbf{I}$ from $\mathcal{D}^{img}$ are input into the 2D detector, which is supervised by projected 3D boxes and the corresponding class, denoted as $\{K \times \mathbf{b}_{3D}, \mathbf{c}_{3D}\}$. It is noteworthy to mention that the images $\mathbf{I}$ from from ImageNet1K $\mathcal{D}^{ign}$ are

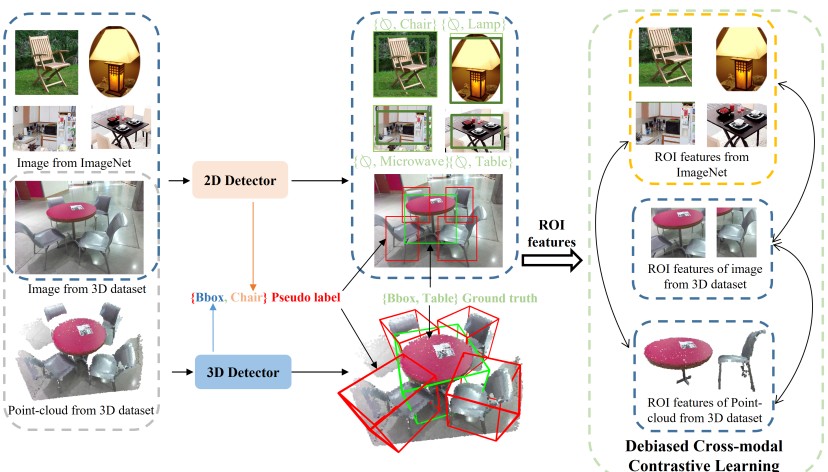

Figure 2: Overview of the phase 2 of OS-3DETIC, which is the core and a synergy of two components: 1) We take advantage of two modalities, image modality for classification and point-cloud modality for localization, to generate pseudo-labels for unseen classes, and 2) we design a de-biased cross-modal contrastive learning to transfer the knowledge from image to point-cloud. Note that before phase 2, we first train both the 2D detector and the 3D detector as similar to Detic Zhou et al. (2022). The green brackets denote ground truth of seen classes. For ImageNet1K, the "bbox" is not provided, so denoted as $\varnothing$. The red brackets denote pseudo-label, where bounding boxes come from the output of the 3D detector, and classes come from the output of the 2D detector.

also input into the same 2D detector with only image-level labels (category) are provided. Following Detic Zhou et al. (2022), we choose the max-size proposal $\mathbf{f}_{maxsize}$ and apply classification label $\mathbf{c}_{2D}^{ign}$ to supervised the classifier in the 2D detector. The loss in the phase 1 training is given by

$$
\begin{aligned}
L^{phase1} =& L_{box}^{3D}(\mathbf{b}_{3D}, \hat{\mathbf{b}}_{3D}) + L_{cls}^{3D}(\mathbf{c}_{3D}, W_{3D}\mathbf{f}_{3D}) + \\
& L_{box}^{2D}(K \times \mathbf{b}_{3D}, \hat{\mathbf{b}}_{2D}) + L_{cls}^{2D}(\mathbf{c}_{3D}, W_{2D}\mathbf{f}_{2D}) + L_{cls}^{ign}(\mathbf{c}_{2D}^{ign}, W_{2D}\mathbf{f}_{maxsize}),
\end{aligned}
\tag{1}
$$

where $L_{box}^{3D}$ and $L_{box}^{2D}$ follow Misra et al. (2021) and Carion et al. (2020). $L_{cls}^{3D}$ and $L_{cls}^{2D}$ are cross-entropy loss for classification. $W_{3D}$ and $W_{2D}$ denote the classifier in the 3D detector and the 2D detector, respectively.

In phase 2, we first generate pseudo-labels for unseen classes. The pseudo-labels contain two parts: bounding box from the 3D detector, and class from the 2D detector. Specifically, since we already exploit ImageNet1K to train the 2D detector in the phase 1, as similar to Detic Zhou et al. (2022), the 2D detector is able to classify unseen classes so that we can use its classification results as relatively accurate pseudo-labels. To leverage this, we crop the image region of the projected 3D detection and use the 2D detector's classifier to generate its class label. For the bounding box pseudo-labels, we take advantage of the generalizability of localization for the point-cloud detector, as mentioned in Fig. 1(b). Besides using pseudo-labels, we also design a de-biased cross-modal contrastive learning to better transfer the knowledge from 2D modality to point-cloud modality. Note that there is significant synergy between this pseudo-label strategy and our proposed de-biased cross-modal contrastive learning. The pseudo-label is beneficial for true positive sampling of cross-modal contrastive learning, and cross-modal contrastive learning transfer the knowledge from image to point-cloud gradually, in turn helping to generate better pseudo-labels. Thus the pseudo-labels are iteratively updated to be of higher quality. Overall, the total loss in phase 2 is given by

$$
\begin{aligned}
L^{phase2} =& L_{box}^{3D}(\bar{\mathbf{b}}_{3D}, \hat{\mathbf{b}}_{3D}) + L_{cls}^{3D}(\bar{\mathbf{c}}_{3D}, W_{3D}\mathbf{f}_{3D}) + L_{box}^{2D}(K \times \bar{\mathbf{b}}_{3D}, \hat{\mathbf{b}}_{2D}) + \\
& L_{cls}^{2D}(\bar{\mathbf{c}}_{3D}, W_{2D}\mathbf{f}_{2D}) + L_{cls}^{ign}(\mathbf{c}_{2D}^{ign}, W_{2D}\mathbf{f}_{maxsize}) + L_{DECC},
\end{aligned}
\tag{2}
$$

where $\bar{\mathbf{b}}_{3D}$ and $\bar{\mathbf{c}}_{3D}$ come from either ground truth or pseudo-labels, and $L_{DECC}$ denotes the loss function of the de-biased cross-modal contrastive learning.

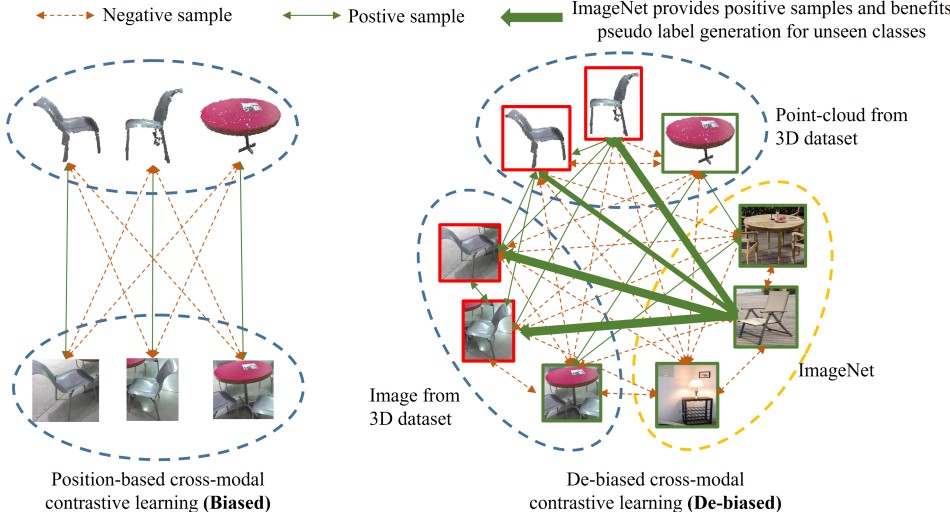

Figure 3: Typical biased cross-modal contrastive learning (left) and the proposed de-biased cross-modal contrastive learning (right). Position-based cross-modal contrastive learning follows the position correspondence with one-to-one match. This may result in assigning inaccurate negative sample. The proposed de-biased cross-contrastive learning (DECC) leverages pseudo-labels to facilitate the biased issues. The red outline denotes it is an unseen class and we assign a pseudo-label to it, and the green outline denotes the seen class with ground truth.

### 3.3 DE-BIASED CROSS-MODAL CONTRASTIVE LEARNING

Before we in-depth illustrate the proposed de-biased cross-modal contrastive learning, we first discuss the typical biased contrastive learning in the following. Cross-modal contrastive learning has been demonstrated as a powerful method used for knowledge transferring Jiang et al. (2019); Radford et al. (2021); Minderer et al. (2022), and enabling zero-shot/open vocabulary classification and detection. Using instance discrimination as the pretext task, typical contrastive learning Radford et al. (2021); Chen et al. (2020); He et al. (2020); Khosla et al. (2020) pulls the semantically-close neighbors together and pushes away non-neighbors. In previous works Radford et al. (2021); Minderer et al. (2022), dissimilar (negative) points are typically taken to be randomly sampled datapoints, implicitly accepting that these points may actually express the same underlying concepts. For example, CLIP Radford et al. (2021) conducts contrastive learning on coupled text and image, while the text caption may describe a bunch of images that have similar semantics. This issue is studied as noisy matching Wu et al. (2021) or biased contrastive learning Chuang et al. (2020). In the image-point-cloud field, existing studies mainly focused on matching the features from the same 3D positions Hou et al. (2021); Liu et al. (2021b); Park et al. (2022). This results in biases, as shown in left part of Fig. 3. We can observe that both within and across modalities, some objects with the same classes are inaccurately assigned as negative samples.

In order to mitigate the biased issue in contrastive learning, we propose a de-biased cross-modal contrastive learning (DECC), as shown in right part of Fig. 3. We take the ROI features $\mathbf{f}_{3D}$ from the 3D detector and $\mathbf{f}_{2D}$ from the 2D detector, and use the ground truth or pseudo-labels $\bar{\mathbf{c}}_{3D}$ to assign the positive and negative samples. For a mini-batch of $\mathbf{f}_{3D}$ and $\mathbf{f}_{2D}$ with total $M$ features from two modalities, we further use a linear projection to transform these feature into hidden representations, which is denoted as $h_i$, where $i = 1, 2, 3, ..., M$. Then the loss is given by

$$L_{DECC} = -\frac{1}{M} \sum_{i=1}^{M} \log \frac{\sum_{t=0}^{m} e^{h_i^\top h_t / \tau(dist_{it})}}{\sum_{j=0}^{M} e^{h_i^\top h_j / \tau_0}}, \quad (3)$$

where $m$ is the number of positive sample corresponding to $h_i$ ($m <= M$), $\tau_0$ is the base temperature. It is noteworthy to mention that different from previous contrastive learning with constant temperature $\tau$ for each sample, we here adopt a distance-aware temperature $\tau(dist_{ij}) = \tau_0 \times \gamma^{dist_{ij}}$ for positive sample, where $\gamma$ are hyperparameters. In particular, we calculate the Euclidean distance

of the two samples in 3D space. Note that if the one of the sample from ImageNet, we consider the distance between ImageNet and any of other sample as 1. Intuitively, the correlation between two close-by objects is greater than that of two distant object, thus we scale the temperature with distance-aware strategy to facilitate this correlation. Specifically, If $\gamma < 1.0$, then $L_{DECC}$ pays more attention to connect close-by objects, and if $\gamma = 1.0$, $L_{DECC}$ degenerates to de-biased cross-modal contrastive learning with constant temperature, while if $\gamma < 1.0$, $L_{DECC}$ focuses on bridge two distant objects. In practices, we set $\gamma > 1.0$ is to enforce the alignment of close-by objects.

## 4 EXPERIMENTS

In this section, we compare the proposed OS-3DETIC with popular baselines on two widely used 3D detection datasets, SUN RGB-D Song et al. (2015) and ScanNetDai et al. (2017). Then we conduct sufficient analysis and ablation studies to explore why OS-3DETIC works. The details are illustrated in the below.

### 4.1 DATASETS AND EVALUATION METRIC

**SUN RGB-D** Song et al. (2015) and **ScanNetV2** Dai et al. (2017) are two widely used 3D detection datasets. We follow the data preprocessing procedure from VoteNet Qi et al. (2019b), except for the sampling of unseen classes. We randomly select 10 unseen classes for both two datasets. Specifically, toilet, bed, chair, bathtub, sofa, dresser, scanner, fridge, lamp and desk are unseen classes for SUN RGB-D, while table, night stand, cabinet, counter, garbage bin, bookshelf, pillow, microwave, sink and stool are seen classes. toilet, bed, chair, sofa, dresser, table, cabinet, bookshelf and pillow are unseen classes for ScanNet, and bathtub, fridge, desk, night stand, counter, door, curtain, box, lamp and bag are seen classes.

The metrics we use in the experiments are Average Precision (AP), mean Average Precision (mAP) and Average Recall (AR) at IoU thresholds of 0.25, denotes as $AP_{25}$, $mAP_{25}$, $AR_{25}$, respectively. *mAP takes both classification and localization in to consideration, the large the mAP means the better the detection, whereas AR mainly focuses on localization. Therefore, compared to AR, mAP is a better metric for evaluating detection result.*

### 4.2 EXPERIMENTAL SETUP AND IMPLEMENTATION DETAILS

**Implementation** The proposed OS-3DETIC is mainly based on 3DETR Misra et al. (2021), yet it can also generalize to other point-cloud detectors. During training, a minibatch of data consists of the point-cloud with its paired image and several images that from ImageNet. 3DETR as the 3D detector consumes the point-cloud, and DETR Carion et al. (2020) as the 2D detector deals with images. Note that only classification loss is used in the DETR when the input is images from ImageNet. We select the max-size ROI features detected by DETR and apply the classfication loss on it, as following Detic Zhou et al. (2022). $\tau_0$ and $\gamma$ are two hyperparameters in the distance-aware temperature strategy, in practice, $\tau_0$ is set to 0.2, $\gamma$ is set to 1.1, and the distance between image from ImageNet and any other sample is 1.0. For both 3DETR and DETR, a unified learning rate is set to $2 \times 10^{-5}$, with batch size of 4 for each GPU, and we train our model on 8 RTX 2080 TI gpus. We train 200 epochs for the phase 1, and 200 epochs for the phase 2.

**Pseudo-Label Generation** Pseudo-label consists of two elements: 3D bounding box and category. We first generate initial pseudo-labels after finishing phase 1, and update them iteratively. As we discussed in Section 3.2, the 3D detector performs generalizable object localization, while the 2D detector is trained on the ImageNet dataset with sufficient knowledge. Therefore, the bounding box comes from the 3D detector and the class label comes from the 2D detector. Furthermore, we only keep proposals with high confidence and filter out the proposals that are duplicated with ground truth or no point in it. Finally, only top-$k$ proposals are kept for each unseen class, and $k$ increases linearly every 50 epochs in phase 2. $k$ is default as 50, and increases 10 (default) every 50 epochs.

Table 1: Detection results ($AP_{25}$) on unseen classes of SUN RGB-D.

| Method | toilet | bed | chair | bathtab | sofa | dresser | scanner | fridge | lamp | desk | mean |
|---|---|---|---|---|---|---|---|---|---|---|---|
| GroupFree3D Liu et al. (2021d) | 0.23 | 0.04 | 1.25 | 0.03 | 0.21 | 0.21 | 0.14 | 0.10 | 0.03 | 3.02 | 0.53 |
| VoteNet Qi et al. (2019a) | 0.12 | 0.05 | 1.12 | 0.03 | 0.09 | 0.15 | 0.06 | 0.11 | 0.04 | 2.10 | 0.39 |
| H3DNet Zhang et al. (2020) | 0.24 | 0.10 | 1.28 | 0.05 | 0.22 | 0.22 | 0.13 | 0.14 | 0.03 | 6.09 | 0.85 |
| 3DETR Misra et al. (2021) | 1.57 | 0.23 | 0.77 | 0.24 | 0.04 | 0.61 | **0.32** | 0.36 | 0.01 | 8.92 | 1.31 |
| OS-PointCLIP Zhang et al. (2021) | 7.90 | 2.84 | **3.28** | 0.14 | 1.18 | 0.39 | 0.14 | 0.98 | **0.31** | 5.46 | 2.26 |
| OS-Image2Point Xu et al. (2021a) | 2.14 | 0.09 | 3.25 | 0.01 | 0.15 | 0.55 | 0.04 | 0.27 | 0.02 | 5.48 | 1.20 |
| Detic-ModelNet Zhou et al. (2022) | 3.56 | 1.25 | 2.98 | 0.02 | 1.02 | 0.42 | 0.03 | 0.63 | 0.12 | 5.13 | 1.52 |
| Detic-ImageNet Zhou et al. (2022) | 0.01 | 0.02 | 0.19 | 0.00 | 0.00 | 1.19 | 0.23 | 0.19 | 0.00 | 7.23 | 0.91 |
| Ours | **43.97** | **6.17** | 0.89 | **45.75** | **2.26** | **8.22** | 0.02 | **8.32** | 0.07 | **14.60** | **13.03** |
| Improvement | +36.07 | +3.33 | -2.39 | +45.51 | +1.08 | +7.03 | -0.30 | +7.34 | -0.24 | +5.68 | +10.77 |

Table 2: Detection results ($AP_{25}$) on unseen classes of ScanNet.

| Method | toilet | bed | chair | sofa | dresser | table | cabinet | bookshelf | pillow | sink | mean |
|---|---|---|---|---|---|---|---|---|---|---|---|
| GroupFree3D Liu et al. (2021d) | 0.63 | 0.52 | 1.52 | 0.52 | 0.20 | 0.59 | 0.52 | 0.25 | 0.01 | 0.15 | 0.49 |
| VoteNet Qi et al. (2019a) | 0.04 | 0.02 | 0.12 | 0.00 | 0.02 | 0.11 | 0.07 | 0.05 | 0.00 | 0.00 | 0.04 |
| H3DNet Zhang et al. (2020) | 0.55 | 0.29 | 1.70 | 0.31 | 0.18 | 0.76 | 0.49 | 0.40 | 0.01 | 0.10 | 0.48 |
| 3DETR Misra et al. (2021) | 2.60 | 0.81 | 0.90 | 1.27 | 0.36 | 1.37 | 0.99 | 2.25 | 0.00 | 0.59 | 1.11 |
| OS-PointCLIP Zhang et al. (2021) | 6.55 | 2.29 | 6.31 | 3.88 | 0.66 | 7.17 | 0.68 | 2.05 | 0.55 | 0.79 | 3.09 |
| OS-Image2Point Xu et al. (2021a) | 0.24 | 0.77 | 0.96 | 1.39 | 0.24 | 2.82 | 0.95 | 0.91 | 0.00 | 0.08 | 0.84 |
| Detic-ModelNet Zhou et al. (2022) | 4.25 | 0.98 | 4.56 | 1.20 | 0.21 | 3.21 | 0.56 | 1.25 | 0.00 | 0.65 | 1.69 |
| Detic-ImageNet Zhou et al. (2022) | 0.04 | 0.01 | 0.16 | 0.01 | 0.52 | 1.79 | 0.54 | 0.28 | 0.04 | 0.70 | 0.41 |
| Ours | **48.99** | **2.63** | **7.27** | **18.64** | **2.77** | **14.34** | **2.35** | **4.54** | **3.93** | **21.08** | **12.65** |
| Improvement | +42.44 | +0.34 | +0.96 | +14.76 | +2.11 | +7.17 | +1.36 | +2.29 | +3.38 | +20.29 | +9.56 |

## 4.3 MAIN RESULTS

As there is no baseline directly solving the problem of both open-set 3D point-cloud localization and classification, we mainly compare OS-3DETIC with state-of-the-art 3D point-cloud detectors Liu et al. (2021d); Qi et al. (2019a); Zhang et al. (2020); Misra et al. (2021) and some well-known works Zhang et al. (2021); Xu et al. (2021a); Zhou et al. (2022) that study either transferability in point-cloud or 2D open-set detection. Specifically, the baselines we use include:

- GroupFree3D Liu et al. (2021d), VoteNet Qi et al. (2019a), H3DNet Zhang et al. (2020), 3DETR Misra et al. (2021) are well-known and representative 3D point-cloud detectors that are chosen as our baselines. Specifically, these four baselines are trained on the seen classes while being tested on the unseen.

- The second is PointCLIP Zhang et al. (2021) which bridges the point-cloud and text domain. We use it directly as a pre-trained open-set 3D classifier and replace the classifier of 3DETR with PointCLIP. This baseline is denoted as **OS-PointCLIP**, which is similar to well-known 2D open-set detection works Bansal et al. (2018); Gu et al. (2021); Zhou et al. (2022) that replace the classifier in the detector with the generalizable classifier.

- Besides, Xu et al.Xu et al. (2021a) transfer the image pre-trained transformer to the point-cloud by copying or inflating the weights. Similarly, we copy the weights of the transformer and the classifier from pre-trained DETR (pre-trained on COCO Lin et al. (2014)) to 3DETR and finetune the set aggregation module and the 3D box head. We term this baseline as **OS-Image2Point**.

- Moreover, Detic Zhou et al. (2022) leverages large-scale classification dataset (ImageNet) to broaden the 2D detector, here we directly extend the idea to 3D open-set detection. Specifically, we consider two manners, extend the classifier via ModelNet or ImageNet, and term them as **Detic-ModelNet** and **Detic-ImageNet**, respectively.

The results are presented in Tables 1 and 2. We can observe that **OS-PointCLIP** achieves $mAP_{25}$ of 2.26% and 3.09%, outperforming the other baselines on both SUN RGB-D and ScanNet. Furthermore, **Detic-ModelNet** reaches $mAP_{25}$ of 1.52% and 1.69% on SUN RGB-D and ScanNet, respectively. It can be observed that both **OS-PointCLIP** and **Detic-ModelNet** outperforms **OS-Image2Point** and **Detic-ImageNet**. Indeed, **OS-PointCLIP** and **Detic-ModelNet** enhances the classifier of the detector via introducing the pretraining on ModelNet, which is a a 3D classification

Table 3: Ablation study on different components.

| Baseline | Pseudo-Label | Augmentation Based CL | Position Based CL | Class Based CL | Distance-Aware Temperature | $mAP_{25}$ | $AR_{25}$ |
|:---:|:---:|:---:|:---:|:---:|:---:|:---:|:---:|
| ✓ | | | | | | 1.31 | 27.00 |
| ✓ | ✓ | | | | | 12.55 | 38.35 |
| ✓ | ✓ | ✓ | | | | 10.48 | 36.44 |
| ✓ | ✓ | | ✓ | | | 12.36 | 37.35 |
| ✓ | ✓ | | | ✓ | | 12.92 | **42.48** |
| ✓ | ✓ | | | ✓ | ✓ | **13.03** | 37.71 |

(a) Performance vs. Data Ratio.

(b) Performance vs. Iteration.

Figure 4: (a). illustrates relation between $mAP_{25}$, $AR_{25}$ and the data regime of both 2D and 3D detection, the green, orange, solid and dashed lines represent point-cloud, image, $mAP_{25}$ and $AR_{25}$, respectively. (b). illustrates the relation between $mAP_{25}$, $AR_{25}$ and pseudo-label iteration, the blue and red lines represents $mAP_{25}$ and $AR_{25}$, respectively.

dataset, while **OS-Image2Point** and **Detic-ImageNet** try to transfer knowledge from image (COCO and ImageNet) to point-cloud. This contrast shows that the modality gap between 2D and 3D does indeed hinder knowledge transfer. The results also demonstrate that directly plugging and playing the Detic method on 3DETR with introducing ImageNet is infeasible to transferring the knowledge from 2D image to 3D. Nonetheless, our method achieves $mAP_{25}$ of 13.03% and 12.65% on SUN RGB-D and ScanNet, respectively, which proves the proposed OS-3DETIC can indeed make use of the image knowledge, to achieve open-set 3D detection.

### 4.4 ANALYSIS AND ABLATION STUDY

**Ablation Study on Different Components**   We conduct an ablation study on SUN RGB-D, the results of unseen classes are reported in Table 3. Our baseline is 3DETR which is trained only on seen categories. "Pseudo-Label" denotes to train with pseudo-label. "Augmentation Based CL" denotes we follow SimCLR Chen et al. (2020); He et al. (2020) that utilize data augmentation strategy in contrastive learning, which however, is a biased contrastive learning setting as well. Position-based contrastive learning takes only the same instance from the paired image/point-cloud as positive. Class based contrasitve learning takes positive so long as two samples belong to the same category. "Distance-Aware Temperature" refer to the strategy that is proposed in Section 3.3. We can observe that, first, pseudo-label brings the largest improvement, actually, pseudo-label implicitly transfers the knowledge contained in ImageNet to a totally different modality, the point-cloud modality, via providing useful class labels. This not only validates our hypothesis that the 3D detector functions as a general region proposal network, but it also demonstrates the effectiveness of introducing large-scale image-level supervision for classification. Second, class-based contrastive learning and distance-aware temperature further significantly improve the performance, while traditional augmentation-based contrastive learning and position-based contrastive learning hurt the performance, which indicates the weakness of biased contrastive learning, and demonstrates the proposed de-biased cross-modal contrastive learning indeed benefits the point-cloud detector to learn general representations.

**Analysis of Performance on Different Training Data Ratio**   Fig. 4(a) illustrate relation between $mAP_{25}$, $AR_{25}$ and the different training data ratio. the left $y$ axis denotes $mAP_{25}$, the right $y$ axis

denotes $AR_{25}$, and the $x$ axis represents the data ratio that used during training. We draw both $mAP_{25}$, $AR_{25}$ results of the point-cloud and the paired image branch. On one hand, compared with $mAP_{25}$, $AR_{25}$ converges with relatively less training data, which exists in both the image and the point-cloud branches. $AR_{25}$ partially reflects localization ability, which means we can train a generalizable bounding box detector with few annotations. On the other hand, compared with $AR_{25}$ of image branch, the $AR_{25}$ of point-cloud branch converges with less training data, which means the localization ability of 3D detector are better than 2D detector, verifying the strategy of using the point-cloud detector to generate bounding box pseudo-labels. Moreover, the converged $mAP_{25}$ of the image branch is better than the point-cloud branch. This may be because texture and detail account a lot for classification, which however is the weakness of the point-cloud detector.

**Analysis on Pseudo-Label Effects**   Fig. 4(b) illustrates the relation between $mAP_{25}$, $AR_{25}$ and pseudo-label iteration. During the second phase, we iteratively update the pseudo-label every 50 epochs. The results show that the more the iteration, the better the pseudo-labels should be, thus leading to the better the performance.

### 4.5   QUALITATIVE RESULTS

Fig. 5 provides the qualitative results. We can observe that Baseline is able to predict the relatively accurate locations compared with ground truth, but the size, center, and direction are incorrect, especially the unseen class bounding boxes (chair in Scene 1, sofa in scene 2, bed in scene 3 and sofa in scene 4). Compare baseline with OS-3DETIC, OS-3DETIC performs much better on the unseen classes.

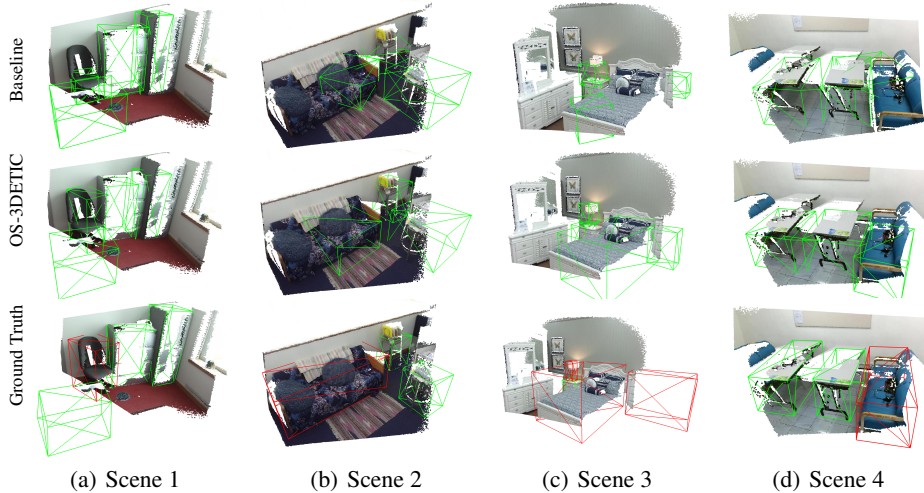

| (a) Scene 1 | (b) Scene 2 | (c) Scene 3 | (d) Scene 4 |

Figure 5: **Visualization of detection result**. 4 column represent 4 different scenes, the comparison conducts among **Baseline**, **OS-3DETIC** and **Ground Truth**. All the bouning boxes in Baseline and OS-3DETIC are predicted bounding boxes, And the red bounding boxes in Ground Truth represent unseen classes samples, and the green represent seen classes.

## 5   CONCLUSION

In this paper, we study a new problem of open-set 3D detection. The proposed method OS-3DETIC introduces ImageNet1K to help open-set point-cloud detection. OS-3DETIC consists of two components: 1) we take advantage of two modalities — the image modality for classification and the point-cloud modality for localization, to generate pseudo-labels for unseen classes, and 2) de-biased cross-modal contrastive learning to transfer the knowledge from images to point-clouds. Extensive experiments show that we improve a wide range of baselines by a large margin, demonstrating the effectiveness of the proposed method. We also provide explanations of why it works via ablation studies and analyzing the representations. We do hope our work could inspire the research community to further explore this field.

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
