# OpenReview forum: "Open-Set 3D Detection via Image-level Class and Debiased Cross-modal Contrastive Learning"
_ICLR.cc/2023/Conference — Submitted to ICLR 2023_

### Official Review · Reviewer_2qoz · 2022-10-24

**Confidence:** 4
**Correctness:** 3
**Technical Novelty And Significance:** 3
**Empirical Novelty And Significance:** 3
**Recommendation:** 6

**Clarity, Quality, Novelty And Reproducibility:**

The paper introduced a new open-set 3D object detection problem by cross-modal learning. Through technical novelty in the individual parts are limited, the overall framework seem to be interesting and perform well in provided experiments. Most of description seems to clear and be detailed for reproducing the proposed method.

**Strength And Weaknesses:**

[+] First of all, the motivation of the paper seems to be meaningful and pragmatic for 3D object detection in the perspective of the generalized 3D object detection and difficulty of the 3D annotation procedure. The key idea is very intuitive how to obtain the pseudo label for 3D object detection and the overall framework seems to be well-designed to embody the author’s purpose. Even if the technical novelties is limited, the idea of de-bias contrastive learning approach seems to be reasonable for overcoming the limitation of position-based contrastive learning, which is a hard semantic relationship between defined pairs.

[-] One concern is an influence of de-bias learning to 3D localization. Unlike 2D localization problems, the orientation is included in 3D localization task. The orientation is known to bias to the part or certain positions of the object. As mentioned in the paper, if the ROI features is used for de-bias learning, it might conflict with the problem to estimate the orientation. I pretty agree that debias learning is helpful to recognize the class, but I am not sure that this learning strategy is also beneficial to 3D localization task.

[-] According to Table.3, the authors argued that the 3D detector can be seen as general region proposal network. In other words, the proposed method can be highly affected by the quality of localization results from 3D detection model. Compared with images, point-cloud has many object shapes that are difficult to understand and define to the object due to occlusion, truncation etc. Moreover, since 3D object detection estimates the abstract scale of the object, the scale of the box could be biased for a certain class. Therefore, I doubt whether the 3D detector can play the same role as RPN of Faster-RCNN, which is trained by various scaled boxes and appearance or 2D object proposal methods.


**Summary Of The Paper:**

In this paper, the authors addressed an open-set 3D detection problem to broaden the vocabulary of the point-cloud detection without laborious and expensive data annotation. Inspired by previous open-set works, the authors proposed OS-3DETIC consisting of two main functions as classification using image-based model and localization using point-cloud model. Furthermore, to maximize the capability of transferring knowledge between two modalities, the authors proposed a de-biased cross-modal contrastive learning as an auxiliary task in the open-set detection problem. In experimental section, the proposed method showed the significant improvement in 3D unseen benchmark and is validated by various ablation studied provided.

**Summary Of The Review:**

I mentioned all comments including reasons and suggestions in the above sections. I recommend that the author will provide all the concerns, and improve the completeness of the paper.

---

> ### Author Response · Authors · 2022-11-19
> **[2/2] Response to Reviewer 2qoz**
>
> ***Q3: Whether the 3D detector is able to propose 3D bounding box for unseen classes just like RPN does?***
>
> > *&emsp;We do agree with you that 3D detection on point clouds is a non-trivial task, even for humans, due to occlusion and truncation. However, the experimental results do not match our expectations, which demonstrates that it is possible to localize 3D unseen objects. Specifically, as we can see in the following two experiments:*
> >
> > &emsp;
> >
> > 1. *As we can see in Table 3, our baseline directly trained on seen classes yields an average recall of 27% for the unseen categories, which means at least 27% of the unseen objects are localized. In other words, the 3D detector is able to localize unseen objects even though it is trained on seen classes.*
> > 2. *As we can see in Figure 1 (b), 1% training labels of ScanNet point-cloud yield an average recall of 28.93%, and the average recall achieved by 10% training labels is almost on par with 100% training label, which further demonstrates localization ability is easy to learn ([1] has the same observation), even though it is trained on a small amount of training label.*
> >
> > &emsp;
> >
> > &emsp;*Both above experiments indicate that the 3D detector indeed able to propose 3D bounding boxes for unseen classes. In fact,  3D detection can be decoupled into localization and classification, and the aforementioned occlusion and truncation (incomplete point-cloud) make classification become challenging, yet we argue that localization is less affected (a naive solution is to first cluster the point-cloud, and then compute a box for each clustered points group, not to mention we have bounding box of seen classes to learn from). In other words, a comprehensive understanding of a scene is required for classification, but for localization, even an unsupervised approach is able to propose some valid 3D bounding boxes based on geometry cue (for example: car is always above the ground).*
> >
> > &emsp;
> >
> > *&emsp;In addition, compared to 2D image that consists of RGB pixels, point-cloud consists of a series of points, where each point is represented by xyz location which is inherently suitable for localization.*
> >
> > &emsp;
> >
> > *&emsp;Therefore, we argue that even if occlusion and truncation make classification difficult,  3D detectors are still able to propose 3D bounding boxes as RPN does.*
> >
> > &emsp;
> >
> > ------
> >
> > > [1] Semantic Abstraction: Open-World 3D Scene Understanding from 2D Vision-Language Models. *CoRL*. 2022.
> >
> > ------
>
> ------

---

> ### Author Response · Authors · 2022-11-19
> **[1/2] Response to Reviewer 2qoz**
>
> *&emsp;We sincerely thank you for the efforts of reviewing our work. After reading your comments carefully, we summarize the following 3 questions. We hope these answers can address your concerns. If you have other concerns, we will reply as soon as possible.*
>
> &emsp;
>
> ***Q1: Whether the so-called ROI feature is able to estimate the orientation?***
>
> > &emsp;*We term the feature corresponding to each prediction as the ROI feature for intuitive description. In fact, since 3DETR is a transformer-based detector, the receptive field of ROI features is the entire point cloud. Therefore, the feature of each prediction actually sees points outside the predicted 3D bounding box and knows where it is in terms of the entire point-cloud. In other words, the ROI feature is able to fit the biased orientation distribution.*
> >
> > ------
>
> ------
>
> &emsp;
>
> ***Q2: Whether the proposed debiased cross-modal contrastive learning is beneficial to localization?***
>
> > &emsp;*The de-biased contrastive learning is beneficial to localization. Our evidence is as follows:*
> >
> > &emsp;
> >
> > - *Intuitively, the spirit of de-biased cross-modal contrastive learning is to transfer the knowledge from 2D (ImageNet) to 3D point-cloud. And the models trained on 2D datasets see much more diverse samples than it does in 3D. In other words, 2D image features are more generalizable than 3D point-cloud features. Therefore, the de-biased cross-modal contrastive help point-cloud detector learns a strong generalization ability feature from image to point-cloud.*
> >
> >   &emsp;
> >
> > - *Moreover, as we can see in Table 3, the class based contrastive learning (which is belong to de-biased contrastive learning) improves Average Recall from 27.0% to 42.48%, which means that 42.48% of the ground truth objects are detected, compared to the baseline trained with only the ground truth of the seen category. Namely, the proposed de-biased cross-modal contrastive learning is beneficial for 3D detection.*
> >
> >   ------
>
> ------

---

> ### Author Response · Authors · 2022-12-06
> **Looking forward to hearing from you.**
>
> *Dear Reviewer 2qoz,*
>
> *&emsp; We would like to thank you again for your time and effort in reviewing our work. As the deadline for discussion approaches, we want to follow up to see if our responses address your concerns. We would be very grateful to hear additional feedback from you and would be happy to provide further clarification if needed.*
>
> *&emsp;Thank you again for your time and effort.  Looking forward to hearing from you.*
>
> *&emsp;*
>
> *Best,*
>
> *Authors of paper 5723*

---

> ### Author Response · Authors · 2022-12-13
> **Final reply and thank you for your efforts**
>
> *Dear Reviewer 2qoz,*
>
> *&emsp; Thank you very much for your valuable time and comments. The discussion phase is coming to an end. We are glad our responses address your concerns. We have also uploaded the revision of the paper accourding to your suggestions. We sincerely appreciate if you could consider to update the rating if our responses address your questions.*
>
> *&emsp;Thank you very much!*
>
> *&emsp;*
>
> *Best,*
>
> *Authors of paper 5723*

---

### Official Review · Reviewer_6Lrw · 2022-10-25

**Confidence:** 5
**Correctness:** 4
**Technical Novelty And Significance:** 2
**Empirical Novelty And Significance:** 2
**Recommendation:** 6

**Clarity, Quality, Novelty And Reproducibility:**

The writing is general clear.
I checked codes come with the supplementary materials but found no README files & config files which makes the reproduction of this work hard.

**Strength And Weaknesses:**

Pros:
1. The paper is generally easy to follow and proposed framework could serve as a much better new baseline.
2. The results are strong compared with existing works.

Cons:
1. Some baselines settings are not quite fair. Taking Detic-ModelNet and Detic-ImageNet as examples, it is hard to understand why only the classifier of Detic are extended, as the proposed method itself extended both classifier and regressor from Detic.
2. The technical novelty is somewhat limited. The proposed framework is almost a Detic detector with 2D localizer replaced by 3D localizer. The proposed DECC improves only marginally to its pseudo label baseline as shown in Table 3.

Miscs:
1. Figure 4 has jammed y-axis legend texts for both sub-figures.

[1] Detecting Twenty-thousand Classes using Image-level Supervision, Zhou et al, ECCV 2022


**Summary Of The Paper:**

The authors proposed a framework for performing open vocabulary 3D object detection. The proposed framework mostly follows the 2D open vocabulary detector Detic[1] with 2D localizer replaced by 3D localizer. Different from Detic, the proposed framework introduced a contrastive feature learning module named debiased cross-modal contrastive learning(DECC) between 2D and 3D RoI features. Experiments on ScanNet and SUN RGB-D showed that the proposed method achieved superior results over existing methods and several baselines.

**Summary Of The Review:**

Overall I think the proposed framework, while lacking technical novelty, could serve as a much stronger baseline compared with existing open vocabulary 3D detection methods. I am leaning to recommend for its acceptance.

---

> ### Author Response · Authors · 2022-11-19
> **[2/2] Response to Reviewer 6Lrw**
>
> ***Q3: Why does DECC seem to improve only marginally over its pseudo-label baseline?***
>
> >&emsp;*The numbers in Table 3 show that distance-aware temperature is 0.11% better than class-based contrastive learning. And it seems that the improvement is relatively small.*
> >
> >&emsp;
> >
> >&emsp;*We want to make it clear that since class-based contrastive learning uses pseudo-labels of unseen classes. Therefore, both class-based contrastive learning and distance-aware temperature are de-biased contrastive learning strategies.*
> >
> >&emsp;
> >
> >&emsp;*Compared with the **baseline + pseudo label** setting, de-biased contrastive learning achieves 0.48% improvement in terms of $mAP_{25}$.*
> >
> >&emsp;
> >
> >&emsp;*Besides, as we can see in Figure 4 (b), with the updating of pseudo-labels, the mAP gradually increased. In fact, de-biased contrastive learning contributes to each time of pseudo-label updating. In other words, the gain of de-biased contrastive learning of the previous iterations is merged to the update of the pseudo-label. To decouple the contribution of pseudo-label and de-biased contrastive learning, we disable the de-biased contrastive learning altogether, the results are given as follows:*
> >
> >&emsp;
> >
> >>| Table 5. Ablation on decoupling pseudo-label and DECC |
> >>| :---------------------------------------------------: |
> >>
> >>|     Iteration     |  0   |   1   |   2   |   3   |   4   |
> >>| :---------------: | :--: | :---: | :---: | :---: | :---: |
> >>| Pseudo-label only | 1.31 | 5.86  |  9.1  | 10.45 | 10.86 |
> >>|     OS-3DETIC     | 1.31 | 6.55  | 10.58 | 12.35 | 13.03 |
> >>|    Improvement    |  0   | +0.69 | +1.48 | +1.90 | +2.17 |
> >
> >&emsp;
> >
> >*&emsp;As we can see, de-biased contrastive learning outperforms the pseudo-label counterpart by 0.69%, 1.48%, 1.90%, 2.17% in each iteration, respectively.*
> >
> >------
>
> ------

---

> ### Author Response · Authors · 2022-11-19
> **[1/2] Response to Reviewer 6Lrw**
>
> *&emsp;We sincerely thank you for the efforts of reviewing our work. After reading your comments carefully, we summarize the following 3 points. We hope these answers can address your concerns. If you have other concerns, we will reply as soon as possible.*
>
> &emsp;
>
> ***Q1: Why do we use the Detic-ModelNet and Detic-ImageNet as our baselines?***
>
> >&emsp;*As there is no baseline directly solving this problem in 3D, while Detic is a well-established approach that is able to detect open-set objects in 2D. Therefore, we adapt the basic idea of Detic into 3D open-set point-cloud detection. As is described in [1]:*
> >
> >&emsp;
> >
> >***&emsp;“Detic, which simply trains the classifiers of a detector on image classification data and thus expands the vocabulary of detectors to tens of thousands of concepts”***
> >
> >&emsp;
> >
> > &emsp;*In other words, the spirit of Detic is using classification dataset (ImageNet21k) to broaden the classifier of the 2D detector. And that is why we only extend the classifier in Detic-ImageNet and Detic-ModelNet.*
> >
> >&emsp;
> >
> >&emsp;*Besides, in order to explore which kinds of datasets (2D or 3D classification dataset) is more beneficial to open-set 3D detection, we further compare ImageNet1K with ModelNet. Results show that:*
> >
> >&emsp;
> >
> >1. *Both image and point-cloud classification datasets barely improve the 3D open-set detection, if they are directly used as Detic does, which further motivates us to investigate how to effectively exploit ImageNet to broaden the 3D detector.*
> >2. *Compared to ImageNet1K (2D classification dataset), ModelNet (3D classification dataset) is more beneficial, if they are directly used as Detic does.*
> >
> >&emsp;
> >
> >------
> >
> >> [1] Detecting twenty-thousand classes using image-level supervision. *ECCV*. 2022.
> >
> >------
>
> ------
>
> &emsp;
>
> ***Q2: How do Detic-ModelNet and Detic-ImageNet perform when we extend both classification and localization to these two baselines?***
>
> > *&emsp;To answer this question, we extend the localization ability to **Detic-ModelNet** and **Detic-ImageNet** via enabling the bounding box regression loss of both seen and unseen classes (ground truth bounding boxes are used to compute the regression loss). We denote these two baselines as **Detic-ModelNet-box** and **Detic-ImageNet-box**, respectively. The results are given in the following table:*
> >
> > &emsp;
> >
> > > | Table 1. Detection results ($AP_{25}$) of supplementary baselines on unseen classes of SUN RGB-D |
> > > | :----------------------------------------------------------: |
> > >
> > > |     Baselines      | tiolet | bed  | chair | bathtub | sofa | dresser | scanner | fridge | lamp | desk | mean |
> > > | :----------------: | :----: | :--: | :---: | :-----: | :--: | :-----: | :-----: | :----: | :--: | :--: | :--: |
> > > | Detic-ImageNet-box |  4.18  | 0.08 | 2.69  |  0.12   | 0.08 |  1.08   |  0.06   |  1.2   | 0.08 | 6.97 | 1.65 |
> > > | Detic-ModelNet-box |  4.52  | 3.37 | 3.86  |   0.1   | 1.09 |  4.56   |  0.13   |  0.25  | 0.27 | 2.45 | 2.06 |
> > > |                    |        |      |       |         |      |         |         |        |      |      |      |
> >
> > &emsp;
> >
> > &emsp;*As we can see, even though extending the localization ability, both **Detic-ModelNet-box** and **Detic-ImageNet-box** are marginally improved, further demonstrating directly using ImageNet or ModelNet as Detic does not perform well for open-set 3D detection.*
> >
> > ------
>
> ------

---

> ### Author Response · Authors · 2022-12-06
> **Looking forward to hearing from you.**
>
> *Dear Reviewer 6Lrw,*
>
> *&emsp; We would like to thank you again for your time and effort in reviewing our work. As the deadline for discussion approaches, we want to follow up to see if our responses address your concerns. We would be very grateful to hear additional feedback from you and would be happy to provide further clarification if needed.*
>
> *&emsp;Thank you again for your time and effort.  Looking forward to hearing from you.*
>
> *&emsp;*
>
> *Best,*
>
> *Authors of paper 5723*

---

> ### Author Response · Authors · 2022-12-13
> **Final reply and thank you for your efforts**
>
> *Dear Reviewer 6Lrw,*
>
> *&emsp; Thank you very much for your valuable time and comments. The discussion phase is coming to an end. We are glad our responses address your concerns. We have also uploaded the revision of the paper accourding to your suggestions. We sincerely appreciate if you could consider to update the rating if our responses address your questions.*
>
> *&emsp;Thank you very much!*
>
> *&emsp;*
>
> *Best,*
>
> *Authors of paper 5723*

---

> ### Comment · Reviewer_6Lrw · 2022-12-13
> **Thanks for the clarification**
>
> My concerns about detic and DIEE are both addressed.

---

### Official Review · Reviewer_fcKh · 2022-10-26

**Confidence:** 3
**Correctness:** 3
**Technical Novelty And Significance:** 2
**Empirical Novelty And Significance:** 3
**Recommendation:** 6

**Clarity, Quality, Novelty And Reproducibility:**

The paper is very clear, the supplement contains additional results and code has been provided, so the work is reproducible.

In terms of novelty, the idea to transfer ImageNet1K label knowledge to a 3D detector seems novel. The approach itself is mostly a combination of existing approaches (DETR, DETR3, Detic, Contrastive learning) but the combination makes sense.

Some things could be made clearer:
- Description of phases and related notation is a bit cumbersome.
- A little more clarity on how you generate 3D boxes for unseen classes in phase 2 would be helpful. If you trained a 3D detector classifier head for the seen classes in scene1, do you just take any objects where classification is anything other than 'background'?
- "max-size proposal fmaxsize " is not particularly clear without having the Detic paper context.

There are a few minor language issues:
“Real world owns a cornucopia of classes”
“Or using 2D detection dataset”
“Ote that”



**Strength And Weaknesses:**

Strengths:
+ Seemingly novel use case of transferring 2D classification knowledge to a point cloud 3D detector. The choice of Imagenet1K seems suitable as a source of supervision, as the classes in question are present in Imagenet.
+ The method is shown to ourperform several reasonable baselines on Sun and Scannet for a number of classes, in some cases by significant margins.
+ Suitable ablations and studies of the effects of iterating the process are done.

Potential weaknesses:
- The start of related work section seems missing. It also claims "to the best of our knowledge, there has been no work on open-set 3D object detection". There is some related work that should be cited. For example Cen, J., Yun, P., Cai, J., Wang, M.Y., Liu, M.: Open-set 3d object detection. 3DV, 2021. Or Wong, K., Wang, S., Ren, M., Liang, M., Urtasun, R.: Identifying unknown instances for autonomous driving. In: CoRL. PMLR (2020). These are not quite the same, as they don't transfer class labels from a trained image classifier, however.
- It is unclear how well the method does compared to full supervision. It would help to share how well DETR3 does on the classes when there is actually supervision.
- Ablation shows that the first step of generating pseudolabels (mostly following the Detic paper) yields most of the gains. The application of "class-unbiased" contrastive training, or "distance-aware" temperature, which are technical details of some minor novelty, yield gains that seems quite small.
- Sun RGB-D dataset contains also images, yet the transfer there is done purely on the 3D point cloud data. It would have been interesting to see how well we can do when also using the RGB data.

**Summary Of The Paper:**

The paper proposes a way to transfer semantic information from Imagenet1K to train a 3D detector on point clouds to detect classes for which there are no 3D labels at all.

The method trains a 2D detector (DETR) and a 3D detector (3DETR) in two stages.
1) The 2D and 3D detector are co-trained, in line with Detic (Zhou 2022), with joint losses. The 3D labels (Sun/ScanNet) train the 3D detector, as well as 2D detector on 2D boxes derived from the 3D GT. The Imagenet labels train the DETR classifier head (applied on the largest detected box in the image).
2) In Phase 2, an augmented dataset is created by running 3D detector to find "unseen classes". The 2D crops obtained from the detections are classifier by the 2D detector. Training continues also incorporating this augmented dataset, but adding contrastive training losses for the 2D crops, that take into account ImageNet classification results (crops with same class are considered positives).
The process of generating the augmented dataset is iterated several times, taking more and more positive examples from the classifier.

The method is evaluated on Sun and Scannet 3D datasets on a number of withheld classes.





**Summary Of The Review:**

The application is interesting and novel. The system is a combination of mostly known components, but they are put together reasonably well. Related work section could be improved. I think it's important to show the gap between the OS3-DETIC system and full supervision.

---

> ### Author Response · Authors · 2022-11-19
> **[2/2] Response to Reviewer fcKh**
>
> ***Q3: How to generate pseudo-label in phase 2?***
>
> >*&emsp;During phase 1, DETR and 3DETR are trained with $\mathcal{D}^{pc}$, $\mathcal{D}^{img}$ and $\mathcal{D}^{ign}$. And after training phase 1, the DETR is able to classify 1k categories of ImageNet1K, and the 3DETR is able to localize 3D objects.*
> >
> >&emsp;
> >
> >&emsp;*In phase 2, we first generate pseudo-labels for unseen classes and then transfer the knowledge from 2D image to 3D point-cloud via pseudo-labeling and de-biased cross-modal contrastive learning.*
> >
> >&emsp;
> >
> >&emsp;*Specifically, we first use 3DETR to generate 3D bounding box proposals $\hat{\mathbf{b}}_{3D}$ on the training set, and then project 3D bounding boxes into 2D corners via projection matrix **K**, after that, minimum enclosed 2D bounding boxes are computed based on the projected corners. Then, DETR is used to classify (the max-size detection result is regarded as the classification result) the cropped 2D patches. Up to now, we have recorded the 3D bounding boxes (3DETR prediction) and its corresponding classification label and confidence (DETR prediction). The next step is to select confident predictions as pseudo-labels. In practice, top-k (according to classification confidence of DETR) predictions are kept for each unseen class, and k is linearly increased along with training iteration.*
> >
> >------
>
> ------
>
> &emsp;
>
> ***Q4: Why does the debiased cross-modal contrastive learning seem to contribute marginally to the final performance?***
>
> >&emsp;*The numbers in Table 3 show that distance-aware temperature is 0.11% better than class-based contrastive learning. And it seems that the improvement is relatively small.*
> >
> >&emsp;
> >
> >&emsp;*We want to make it clear that since class-based contrastive learning uses pseudo-labels of unseen classes. Therefore, both class-based contrastive learning and distance-aware temperature are de-biased contrastive learning strategies.*
> >
> >&emsp;
> >
> >&emsp;*Compared with the **baseline + pseudo label** setting, de-biased contrastive learning achieves 0.48% improvement in terms of $mAP_{25}$.*
> >
> >&emsp;
> >
> >&emsp;*Besides, as we can see in Figure 4 (b), with the updating of pseudo-labels, the mAP gradually increased. In fact, de-biased contrastive learning contributes to each time of pseudo-label updating. In other words, the gain of de-biased contrastive learning of the previous iterations is merged to the update of the pseudo-label. To decouple the contribution of pseudo-label and de-biased contrastive learning, we disable the debiased contrastive learning altogether, the results are given as follows:*
> >
> >&emsp;
> >
> >>| Table 5. Ablation on decoupling pseudo-label and DECC |
> >>| :---------------------------------------------------: |
> >>
> >>|     Iteration     |  0   |   1   |   2   |   3   |   4   |
> >>| :---------------: | :--: | :---: | :---: | :---: | :---: |
> >>| Pseudo-label only | 1.31 | 5.86  |  9.1  | 10.45 | 10.86 |
> >>|     OS-3DETIC     | 1.31 | 6.55  | 10.58 | 12.35 | 13.03 |
> >>|    Improvement    |  0   | +0.69 | +1.48 | +1.90 | +2.17 |
> >
> >&emsp;
> >
> >*&emsp;As we can see, de-biased contrastive learning outperforms the pseudo-label counterpart by 0.69%, 1.48%, 1.90%, 2.17% in each iteration, respectively.*
> >
> >------
>
> ------
>
> &emsp;
>
> ***Q5: Missing citation to some related works.***
>
> > *&emsp;Thank you for your suggestion, in fact, due to the page limitation, the discussion and citation of these two related references are presented in the Section A.1 of the appendix. In order to emphasize the difference between OS-3DETIC and existing works, we move this discussion to Section 2 in the revised version.*
> >
> > ------
>
> ------

---

> > ### Comment · Reviewer_fcKh · 2022-12-02
> > **Response to authors**
> >
> > Thank you for the clarifications, they address the questions I raised. Please add more of that information to the final draft.

---

> > > ### Author Response · Authors · 2022-12-03
> > > **Thank you for your constructive feedback**
> > >
> > > *Dear Reviewer fcKh,*
> > >
> > > *&emsp; We appreciate your constructive and useful feedback. It's great to hear that our response has addressed your concerns. As suggested, all the above results have been added to the revised supplementary material.*
> > >
> > > *&emsp;Once again, we thank you for your active engagement and valuable suggestions.*
> > >
> > > *&emsp;*
> > >
> > > *Best,*
> > >
> > > *Authors of paper 5723*

---

> ### Author Response · Authors · 2022-11-19
> **[1/2] Response to Reviewer fcKh**
>
> *&emsp;We sincerely thank you for the efforts of reviewing our work. After reading your comments carefully, we summarize the following 5 points. We hope these answers can address your concerns. If you have other concerns, we will reply as soon as possible.*
>
> &emsp;
>
> ***Q1: How does 3DETR perform in the fully supervised manner?***
>
> > *&emsp;The fully supervised results of SUN RGB-D and ScanNet are as follows:*
> >
> > &emsp;
> >
> > >| Table 1. Fully supervised results ($AP_{25}$) of 3DETR on SUN RGB-D. |
> > >| :----------------------------------------------------------: |
> > >
> > >|   Class   | tiolet |  bed  | chair | bathtub | sofa  | dresser | scanner | fridge | lamp  | desk  | table | stand | cabinet | counter |  bin  | bookshelf | pillow | mocrowave | sink  | stool | mean  |
> > >| :-------: | :----: | :---: | :---: | :-----: | :---: | :-----: | :-----: | :----: | :---: | :---: | :---: | :---: | :-----: | ------- | :---: | :-------: | :----: | :-------: | :---: | :---: | :---: |
> > >| $AP_{25}$ | 89.13  | 82.61 | 66.08 |  76.84  | 57.29 |  26.60  |  13.10  | 24.02  | 25.05 | 28.34 | 49.49 | 60.10 |  17.24  | 27.92   | 45.14 |   29.51   | 20.72  |   9.42    | 32.03 | 13.38 | 39.70 |
> > >|           |        |       |       |         |       |         |         |        |       |       |       |       |         |         |       |           |        |           |       |       |       |
> >
> > &emsp;
> >
> > >| Table 2. Fully supervised results ($AP_{25}$) of 3DETR on ScanNet. |
> > >| :----------------------------------------------------------: |
> > >
> > >|   Class   | tiolet |  bed  | chair | sofa  | dresser | table | cabinet | bookshelf | pillow | sink  | babthtub | refrigerator | desk  | stand | counter | door  | curtain |  box  | lamp  | bag  | mean  |
> > >| :-------: | :----: | :---: | :---: | :---: | :-----: | :---: | :-----: | :-------: | :----: | :---: | :------: | :----------: | :---: | :---: | :-----: | :---: | :-----: | :---: | :---: | :--: | :---: |
> > >| $AP_{25}$ | 91.14  | 64.84 | 67.82 | 68.16 |  35.33  | 53.80 |  38.30  |   43.38   | 44.57  | 62.91 |  77.67   |    43.86     | 52.89 | 52.05 |  30.78  | 44.05 |  37.97  | 12.37 | 17.38 | 9.78 | 47.45 |
> > >|           |        |       |       |       |         |       |         |           |        |       |          |              |       |       |         |       |         |       |       |      |       |
> >
> > &emsp;
> >
> > &emsp;*As we can see in this table, the gap between OS-3DETIC and the fully supervised results indicates that there is still much room for further research on this problem.*
> >
> > ------
>
> ------
>
> &emsp;
>
> ***Q2: How well can we do when RGB color is available?***
>
> >*&emsp;To answer this question, we retrain 1) the fully supervised 3DETR with RGB color, 2) OS-3DETIC with RGB color. The results are given in the table below.*
> >
> >&emsp;
> >
> >> | Table 3. Fully supervised results ($AP_{25}$) of 3DETR with RGB color on ScanNet. |
> >> | :----------------------------------------------------------: |
> >>
> >> |   Class   | tiolet |  bed  | chair | sofa  | dresser | table | cabinet | bookshelf | pillow | sink  | babthtub | refrigerator | desk  | stand | counter | door  | curtain |  box  | lamp  |  bag  | mean  |
> >> | :-------: | :----: | :---: | :---: | :---: | :-----: | :---: | :-----: | :-------: | :----: | :---: | :------: | :----------: | :---: | :---: | :-----: | :---: | :-----: | :---: | :---: | :---: | :---: |
> >> | $AP_{25}$ | 91.02  | 62.49 | 66.62 | 68.00 |  39.37  | 53.19 |  36.79  |   42.02   | 46.05  | 65.98 |  77.11   |    48.20     | 50.41 | 55.54 |  32.63  | 46.80 |  38.03  | 13.76 | 19.54 | 11.62 | 48.26 |
> >> |           |        |       |       |       |         |       |         |           |        |       |          |              |       |       |         |       |         |       |       |       |       |
> >
> >&emsp;
> >
> >> | Table 4. Detection results ($AP_{25}$) of OS-3DETIC with RGB color on the unseen class of ScanNet |
> >> | :----------------------------------------------------------: |
> >>
> >> |   Class   | tiolet | bed  | chair | sofa  | dresser | table | cabinet | bookshelf | pillow | sink  | mean |
> >> | :-------: | :----: | :--: | :---: | :---: | :-----: | :---: | :-----: | :-------: | :----: | :---: | :--: |
> >> | $AP_{25}$ | 50.21  | 2.74 | 7.33  | 18.93 |  2.91   | 14.44 |  2.65   |   4.75    |  4.35  | 22.71 | 13.1 |
> >> |           |        |      |       |       |         |       |         |           |        |       |      |
> >
> >&emsp;
> >
> >*&emsp;As we can see, RGB colors marginally help 3DETR and OS-3DETIC. Intuitively, however, RGB colors are very helpful for human recognition of objects. The reason is that 3DETR is not designed for processing color information, the official version of 3DETR only uses xyz geometry, and OS-3DETIC is mainly based on 3DETR. Nonetheless, finding a way to fully exploit RGB colors is an interesting problem. We will investigate this further in the future.*
> >
> >------
>
> ------

---

> ### Author Response · Authors · 2022-12-13
> **Final reply and thank you for your efforts**
>
> *Dear Reviewer fcKh,*
>
> *&emsp; Thank you for your time and effort. The discussion phase is drawing to a close. We are glad our response addresses your concerns, could you please consider updating your rating to reflect this.*
>
> *&emsp;Thank you again for your valuable suggestions and efforts!*
>
> *&emsp;*
>
> *Best,*
>
> *Authors of paper 5723*

---

### Official Review · Reviewer_rTqP · 2022-10-30

**Confidence:** 5
**Correctness:** 3
**Technical Novelty And Significance:** 3
**Empirical Novelty And Significance:** Not applicable
**Recommendation:** 6

**Clarity, Quality, Novelty And Reproducibility:**

The paper has good clarity and quality. The proposed idea is novel. Given the introduction of the method and the experimental setup, the results are reproducible.

**Strength And Weaknesses:**

Strength:
1. The paper is well motivated to address an important topic in computer vision.

2. The debased cross-modal contrastive learning is interesting.


Weakness:
1. Is the paper trying to localize and assign semantics simultaneously or just assign semantics, by assuming the localization is naturally generalizable to open-set? Please clarify.

2. What is the justification for the statement that the 3D detector is generalizable to unseen categories?

3. How to weigh different terms in Eq. (1) and Eq. (2)?

4. The experiment setup is questionable. The authors propose to randomly select unseen and seen classes, which however unavoidably exists resemblance in semantics and shape. For example, for both SUN-RGBD and ScanNet, semantically similar concepts like desk and table are leaked in unseen categories.

5. Is the unseen / seen category sampling strategy consistent with other referenced open-set methods, e.g., Zhang et al., Xu et al., Zhou et al.?

6. The proposed method underperforms previous works in lamp, scanner and chair on the SUN -RGBD dataset.

7. In the ablation study, distance aware temperature seems to have marginal improvement in mAP but cause significant regression on AP. Please explain further.

8. I would like to see the ablation study on the proposed DECC idea.

9. Writing quality should be much improved. Typos and grammar errors are throughout the manuscript.

**Summary Of The Paper:**

The paper proposes a novel method for open-set 3D detection using image-level class supervision. The core idea is leveraging each of the image and point-cloud modalities to generate pseudo labels for unseen classes. To improve the positive and negative sample matching, the authors propose debiased cross-modal contrastive learning. The proposed approach takes two phases to train, where the first aims to train classification and localization models to generate pseudo labels and the second aims to further facilitate information transferring with the proposed contrastive learning strategy.

The authors have evaluated their method on the SUN RGBD and ScanNet datasets.

**Summary Of The Review:**

Overall, I think the idea of generating pseudo labels and debiased cross-modal contrastive learning is interesting. During the rebuttal, the authors have addressed my concerns. I would like to raise my rating to acceptance.

---

> ### Author Response · Authors · 2022-11-19
> **[2/2] Response to Reviewer rTqP**
>
> ***Q3: What is the relationship between mAP (mean Average precision) and AR (Average Recall), why is the mAP marginally improved, while the AR drops?***
>
> > &emsp;***mAP** takes both **classification** and **localization** in to consideration, the large the **mAP** means the better the detection, whereas **AR** mainly focuses on localization. Therefore, compared to **AR**, **mAP** is a better metric for evaluating detection results.*
> >
> > &emsp;
> >
> > *&emsp;The results in Table 3 show that the Distance-Aware temperature improves mAP while causing AR drops. This phenomenon indicates that the number of both true positives (**correct bounding box**) and false positives (**incorrect bounding box**) are **reduced**, and that the gain from removing false positives is larger than the loss from missing true positives. The reason for the reduction in both true and false positives is that the **non-maximum suppression** module filters out more bounding box predictions when Distance-Aware Temperature is enabled. Distance-Aware Temperature encourages the paired objects in the image and point cloud to be aligned in the embedding space, which is a de-biased strategy to enhance the confidence of the truth-positive prediction and lead to the suppression of other boxes.*
>
> ------
>
> &emsp;
>
> ***Q4: Further analysis and ablation study on the proposed De-biased Cross-Modal Contrastive Learning.***
>
> >*&emsp;Contrastive learning means learning by comparison between similar (positive) and dissimilar (negative) pairs of samples. Specifically, in our case, contrastive learning is used as a bridge to connect and transfer the knowledge from image to point-cloud. However, due to lack of annotation, dissimilar (negative) samples may implicitly accept the false negative samples and result in biased contrastive learning.*
> >
> >&emsp;
> >
> >*&emsp;To address this issue, on the one hand, we first select positive/negative pairs using pseudo labels. On the other hand, we propose the distance-aware temperature to further filter out the potentially false negative samples. Specifically, in practice, γ is set to 1.1, which means that two close objects contribute more to the $L_{DECC}$ loss than two distant objects. In other words, close objects are more likely to come from the same category (e.g. two nearing chairs, pillows, etc.), and distant objects may not.*
> >
> >&emsp;
> >
> >*&emsp;To validate the effectiveness of the proposed de-biased cross-modal contrastive learning, we perform ablation study to compare different contrastive strategies (augmentation-based, position-based, class-based and distance-aware temperature), quantitative results are shown in Table 3 in the main paper, and the qualitative experiments (TSNE visualization) are provided in Figures 7 and 8 in the appendix.*
> >
> >&emsp;
> >
> >*&emsp;To further investigate the influence of distance-aware temperature, we add an ablation study that sets the γ range from 0.5~1.5 (γ is the base of exponential function). The results are shown in the following Table.*
> >
> >&emsp;
> >
> >>| Table1. Ablation on the distance-aware temperature |
> >>| :------------------------------------------------: |
> >>
> >>|          γ |   0.5 |   0.7 |   0.9 |   1.0 |   1.1 |   1.3 |   1.5 |
> >>| ---------: | ----: | ----: | ----: | ----: | ----: | ----: | ----: |
> >>| $mAP_{25}$ | 11.35 | 12.27 | 12.75 | 12.92 | 13.03 | 12.87 | 12.54 |
> >>|            |       |       |       |       |       |       |       |
> >
> >&emsp;
> >
> >*&emsp;As we can see,*
> >
> >&emsp;
> >
> >1. *When γ = 1, the distance-aware temperature degenerates to class-based contrastive learning.*
> >2. *When γ < 1, which means LDECC focusing on distant objects, the mAP drops, compared to γ = 1.*
> >3. *When γ > 1, which means LDECC focusing on close-by objects, the mAP is improved first and then drops, , compared to γ = 1.*
> >
> >&emsp;
> >
> >*&emsp;And the final results show that the OS-3DETIC achieves the best mAP when γ is set to 1.1.*
> >
> >------
>
> ------
>
> &emsp;
>
> ***Q5: How to balance the different terms in the loss function?***
>
> > *&emsp;The balance strategy among detection loss ($L_{box}^{3D}$, $L_{cls}^{3D}$, $L_{box}^{2D}$, $L_{cls}^{2D}$) follows the implementation of 3DETR and DETR. As for the balance between detection loss, ImageNet classification loss, $L_{cls}^{ign}$, de-biased cross-model contrastive loss, $L_{DECC}$, we simply weight these loss values to the same magnitude as the detection loss at the beginning of the training phase 2.*
> >
> > ------
>
> ------

---

> ### Author Response · Authors · 2022-11-19
> **[1/2] Response to Reviewer rTqP**
>
> *&emsp; We sincerely thank you for the efforts of reviewing our work. After carefully reading your comments, we summarize the following 5 points. We hope these answers can address your concerns. If you have other concerns, we will reply as soon as possible.*
>
> &emsp;
>
> ***Q1: Do we deal with classification alone, or with both localization and classification? Why are 3D detectors able to detect unseen categories?***
>
> > *&emsp;The proposed OS-3DETIC is able to handle both localization and classification. Specifically, we observe that localization is relatively easier compared with classification in the point-cloud domain ([1] also have the same finding), which is supported by the following experiments:*
> >
> > &emsp;
> >
> > 1. *As we can see in Table 3, our baseline directly trained on seen classes yields the average recall of 27% for the unseen categories, which means at least 27% of the unseen objects are localized. In other words, the 3D detector is able to localize unseen objects even though it is trained on seen classes.*
> >
> > 2. *As we can see in Figure 1 (b), 1% training labels of ScanNet point-cloud yield average recall of 28.93%, and the average recall achieved by 10% training labels is almost on par with 100% training label, which further demonstrates the localization ability is easy to be learned, even though it is trained on a small amount of training label.*
> >
> >    &emsp;
> >
> > &emsp;*In addition to the generalization ability of the 3D detector, the proposed OS-3DETIC  further enhances the localization of unseen objects, which is supported by the following experiments:*
> >
> > &emsp;
> >
> > 1. *As we can see in Figure 4 (b), with the increase of iteration, average recall of unseen categories is gradually improving, demonstrating the pseudo-label strategy is beneficial for localization.*
> >
> > 2. *As we can see in Table 3, the average recall is at least 9.44% better than the baseline, which further indicates the localization ability can be improved by OS-3DETIC.*
> >
> >    &emsp;
> >
> > *&emsp;Intuitively, introducing ImageNet into our backbone essentially enhances the representation learning of the whole models, which not only improves the ability of learning better embedding for classification, but also delivers distinguished features for localization.*
> >
> > &emsp;
> >
> > *&emsp;In summary, we handle both localization and classification, where the localization capability is initially derived from the generalization capability of the 3D detector and further enhanced by OS-3DETIC.*
> >
> > &emsp;
> >
> > ------
> >
> > > [1] Semantic Abstraction: Open-World 3D Scene Understanding from 2D Vision-Language Models. *CoRL*. 2022.
> >
> > ------
> >
>
> ------
>
> &emsp;
>
> ***Q2: Is it reasonable to randomly select the seen and unseen classes? Is the sampling strategy consistent with other open-set methods?***
>
> >*&emsp;Indeed, in addition to randomly sampling the seen and unseen classes, we can also artificially select seen and unseen classes. However, we argue that random selection is more reasonable.*
> >
> >&emsp;
> >
> >*&emsp;On the one hand, the artificial selection of unseen categories is subjective and biased towards human willingness. On the other hand, to fairly show the results, we conduct multiple experiments by randomly selecting the categories with different random seeds (we provide these results in the appendix, Table 5 - Table 7). Experiments demonstrate that our method is indeed effective with stable improvements over the baselines.*
> >
> >&emsp;
> >
> >*&emsp;Additionally, similar shapes unavoidably exist even if the unseen classes are cherry-picked from the seen ones. Yet we argue that it is not the fault of the random sampling. Actually, similar shapes commonly exist in practical applications since most man-made objects consist of some basic shapes, e.g., cylinders, cuboids, cones, etc. In fact, similar shapes, however, make the 3D detection more challenging, because the detector has to distinguish between these similar objects. As for the example you mentioned (“desk” and “table”), one of the resampling experiments (results are in Table 6 of the appendix) demonstrates that even though both “desk” and “table” are sampled as unseen categories, OS-3DETIC still outperforms the baseline by a large margin.*
> >
> >&emsp;
> >
> >*&emsp;To answer the second question, OS-3DETIC shares the same sampled unseen and seen classes with the other baselines we used. We want to make it clear that these baselines are not directly solving the open-set 3D detection problem, as we discussed in Section 4.3. To compare with these baselines, we extend their ideas to the open-set 3D detection problem, which means the numbers in Tables 1 and 2 are obtained based on our reproduction, where we control for the same sampled classes among all these baselines.*
>
> ------

---

> ### Author Response · Authors · 2022-12-06
> **Looking forward to hearing from you.**
>
> *Dear Reviewer rTqP,*
>
> *&emsp; We would like to thank you again for your time and effort in reviewing our work. As the deadline for discussion approaches, we want to follow up to see if our responses address your concerns. We would be very grateful to hear additional feedback from you and would be happy to provide further clarification if needed.*
>
> *&emsp;Thank you again for your time and effort.  Looking forward to hearing from you.*
>
> *&emsp;*
>
> *Best,*
>
> *Authors of paper 5723*

---

### Decision · Program_Chairs · 2023-01-20

**Decision:**

Reject

**Justification For Why Not Higher Score:**

The related work section is very poor and not really understandable.

Reviewers also had a lot of questions during the review cycle.  While the authors provided convincing responses, many of the clarification points did not seem to have made it into the updated revision.

The copyediting of the paper is just poor.  The paper has:
1. Tons of typos
- Figure 1: ration => ratio,
- Figure 3: Postive => Positive
- "from from" => "from"
2. Grammar errors
- "..are also input into the same 2D detector with only image-level labels (category) are provided"
3. Lots of places where parenthetical citation should be used.    While this is a minor issue that could be fixed in a revision pass, it does impact the readability of this paper.  Currently some sections are just a long list of citations that are not properly parenthesized, and in many places, the name of the method just flows into the citation as well.

I would also recommend the equations be more properly typeset.



**Justification For Why Not Lower Score:**

N/A

**Metareview: Summary, Strengths And Weaknesses:**

Summary: The paper proposes to use ImageNet1K to broaden the vocabulary of 3D object detector to tackle open-set object detection in point clouds.  A two phase training strategy is proposed for training a 2D detector and 3D detector.  In the first phase, the two detector are both trained together, with classification labels ImageNet used as weak supervisory signal (it is used as the label for the largest detected box in the image) for training the 2D detector (following Detic, Misra et al.).  Then in the second phase, the trained 2D detector is used to generate pseudo category labels for bounding boxes detected by the 3D detector for unseen classes.  The two detectors are again trained together, with a de-biased cross-modal contrastive loss.  Experiments on ScanNet and SUN-RGBD show that the proposed method can successfully detect unseen classes.

Strengths:
- The paper investigates the important setting of leveraging 2D data for training 3D object detectors where some classes do not have 3D labeled data
- The proposed method to transferring label knowledge from ImageNet combines prior approaches in a novel and practical way
- The experiments show that proposed approach outperforms baselines

Weaknesses:
- The related work section is poor and hard to read (it consist mostly of a list of citations) without providing much information about the prior work and how this work is different from the prior work.
- The writing for the paper is poor in general.  The paper has a bunch of typos (Figure 1: ration => ratio), some grammar issues, and lots of places where parenthetical citation should be used (e.g. "3DETR Misra et al. (2021)" => "3DETR (Misra et al. 2021)".